# Divergence of cochlear transcriptomics between reference-based and reference-free transcriptome analyses among *Rhinolophus ferrumequinum* populations

Xiaoxiao Shi[1], Jun Li[1], Tong Liu[2], Hanbo Zhao[3], Haixia Leng[1], Keping Sun[1,4]*, Jiang Feng[1,2]*

**1** Jilin Provincial Key Laboratory of Animal Resource Conservation and Utilization, Northeast Normal University, Changchun, Jilin, China, **2** Department of Life Science, Jilin Agricultural University, Changchun, Jilin, China, **3** Agricultural Genomics Institute at Shenzhen, Chinese Academy of Agricultural, Shenzhen, China, **4** Key Laboratory of Vegetation Ecology, Ministry of Education, Changchun, Jilin, China

* sunkp129@nenu.edu.cn (KS); fengj@nenu.edu.cn (JF)

## Abstract

Differences in gene expression within tissues can lead to differences in tissue function. Understanding the transcriptome of a species helps elucidate the molecular mechanisms underlying phenotypic divergence. According to the presence or absence of a reference genome of for a studied species, transcriptome analyses can be divided into reference-based and reference-free methods, respectively. Presently, comparisons of complete transcriptome analysis results between those two methods are still rare. In this study, we compared the cochlear transcriptome analysis results of greater horseshoe bats (*Rhinolophus ferrumequinum*) from three lineages in China with different acoustic phenotypes using reference-based and reference-free methods to explore their differences in subsequent analysis. The results gained by reference-based results had lower false-positive rates and were more accurate because differentially expressed genes among the three populations obtained by this method had greater reliability and a higher annotation rate. Some phenotype-related enrichment terms, including those related to inorganic molecules and proton transmembrane channels, were also obtained only by the reference-based method. However, the reference-based method might have the limitation of incomplete information acquisition. Thus, we believe that a combination of reference-free and reference-based methods is ideal for transcriptome analyses. The results of our study provided a reference for the selection of transcriptome analysis methods in the future.

## Introduction

Variations in gene expression patterns may lead to phenotypic differentiation within and even between species. Understanding the genetic basis of phenotypic differentiation is currently a research hotspot in the field of evolutionary biology [1–3]. With the rise of next-generation

**Data Availability Statement:** Raw data can be obtained in National Center for Biotechnology Information (NCBI) Short Read Archive (SRA)

Database under SRA accession: PRJNA515764. Reference genome can be obtained in NCBI Genome Database under accession: PRJNA489106. All minimal data sets underlying the results described in the paper are in the Supporting Information files.

**Funding:** This work was supported by grant nos. 32171525 and 31961123001 from the National Natural Science Foundation of China (to KS and JF), and grant 20220101291JC from the Natural Science Foundation of Jilin Province (to KS). The funders had no role in study design, data collection and analysis, decision to publish, or preparation of the manuscript.

**Competing interests:** The authors have declared that no competing interests exist.

sequencing techniques, RNA sequencing (RNA-Seq) has been widely used to study gene expression patterns and provides an effective approach for further exploration of the molecular mechanisms underlying phenotypic differences among species [4, 5].

RNA-Seq data can be analyzed with or without a reference genome. For organisms with reference genomes, gene expression patterns can be quantified after detecting differentially expressed genes (DEGs) by mapping filtered sequencing data to annotated reference genomes of a species or its sibling species [6]; for nonmodel organisms without reference genomes, the reference-free method can be used to assemble and annotate transcripts of different lengths to obtain the full length transcript as a reference transcriptome for subsequent research in the absence of a reference genome [7, 8].

Results obtained by the reference-based method could be affected by the accuracy and completeness of the reference genome [6, 9, 10]. Variations of gene-expression patterns among individuals are missed when using only one single reference genome [6]. Because consensus site dinucleotide motifs are used to map reads across splice junctions, genomic variants in the splice site prevents the reads from being mapped to the reference genome, which could result in the incomplete information acquisition [11]. But Lee et al. [12] found that the results of reference-based and reference-free methods had a great consistency in expression level. In contrast, Vijay et al. [13] found that using the reference-based method with reference genomes from distant species (with 15% sequence differences) still helped to obtain more accurate gene expression levels than the reference-free method, even though the transcriptome was well assembled. The reference-free method uses multiple assembly tools and evaluation indicators when assembling a reference transcriptome; thus, the selection of optimal assembly results may vary among different studies [14, 15]. However, most studies focused on generic transcriptome data differences obtained by these two methods, and did not address gene functional differences in subsequent transcriptome analyses.

Echolocation call is an important phenotypic feature of most bats (Chiroptera) that plays an important role in navigation, detection and predation [16–19]. Bats can also use auditory feedback to control vocal frequency [20]. The echolocation acoustic characteristics of bats have an important relationship with their auditory organs [21]. Zhao et al. [22] used the reference-free method to analyze the cochlear transcriptome of three genetic lineages of *Rhinolophus ferrumequinum* in China with different acoustic phenotypes, and found that the DEGs were enriched in neural and learning pathways; those findings indicated that neural activity and learning behavior are related to the variation of echolocation acoustic characteristics of bats. Recently, Jebb et al. [23] released a high-quality complete genome of *R. ferrumequinum*, which provided a good reference genome for transcriptome analysis.

Thus, in this study, we performed the reference-based method using the transcriptome data obtained by Zhao et al. [22] from the cochlea of bats to analyze the DEGs and metabolic pathways, and the relationship between the DEGs and echolocation call variation in bats. Then we compared our results with those of reference-free assembly analysis performed by Zhao et al. [22]. These results will be helpful for understanding the relationship between cochlear gene expression patterns and chiropteran acoustic phenotypes, and provide a reference for the selection of transcriptome analysis methods.

## Materials and methods

### Sample acquisition and information collection

Raw data were obtained from the transcriptome sequences of *R. ferrumequinum* cochlea sequenced by Zhao et al. [22] (obtained from the National Center for Biotechnology Information [NCBI] Short Read Archive [SRA] database under SRA accession: PRJNA515764), which

included 14 individuals from three geographical populations including the northeast genetic lineage (Jilin population, JL01–JL05), central-east lineage (Henan population, HN01–HN05), and southwest lineage (Yunnan population, YN01–YN04) in China. Reference genome and annotation gene model files were downloaded from the NCBI Genome database (accession PRJNA489106).

## Data quality control and reads mapping

To ensure data analysis quality, raw data were filtered and trimmed using fastp v0.19.7 [24]. We removed reads contaminated by adapter, containing more than 15% ploy-N (N means unknown nucleotides) or containing more than 50% low-quality (Qphred $\leq$ 20) bases. At the same time, Q20, Q30 and GC content of the clean data were calculated. We built the index of the reference genome and mapped clean reads to it using Hisat2 v2.0.5 [25].

## Differential expression analysis and DEG comparison

FeatureCounts v1.5.0-p3 was used to count the reads numbers mapped to each gene [26]. The expected values of Fragments Per Kilobase of transcript sequence per Million base pairs sequenced (FPKM) of each gene were calculated based on the gene length of the gene and read counts mapped to this gene. We also calculated the FPKM of each unigene (genes spliced in the reference-free research) to represent gene expression level instead of Reads Per Kilobase per Million mapped reads (RPKM) used in the research [22]. We then performed principal component analysis (PCA) of all individuals using the factoextra v1.0.7 R package using FPKM obtained in the two methods to identify outlier individuals, and removed the outliers, JL2, HN4 and YN3 [22]. Then we repeated PCA to produce the clustering result of the remnant samples. All subsequent analyses were performed excluding those three outlier samples.

Differential expression analysis of remaining individuals between the two population pairs (HN vs. JL, HN vs. YN and YN vs. JL) was performed using the DESeq2 v1.20.0 R package [27], and $p$-values were adjusted using Benjamini and Hochberg correction [28]. Genes with a $p$-adjust value less than 0.05 and absolute value of log2-fold change more than 1 after correction were assigned as DEGs. DEGs obtained in the reference-based method and the reference-free method were recorded separately. The hierarchical clustering heatmap was used to show the DEG expression.

We then compared DEGs obtained by the reference-based method with those obtained by the reference-free method. We first mapped all unigenes to the reference genome using BLASTn v2.11.0 to identify gene sequence locations [29], and the E-value was set to 1E-5. Locations with the longest mapping length were considered the gene locations. We then counted and compared DEGs with annotations obtained by the two methods. And for DEGs obtained by both methods, we performed paired Mann–Whitney U test to compare the gene expression level and gene length of each DEG using the rstatix v0.7.1 R package. Gene expression levels of shared DEGs were represented by lg-(FPKM+1). We also performed GO and KEGG enrichment analyses using clusterProfiler v3.4.4 R package [30]. GO terms and KEGG pathways with FDR value less than 0.05 after FDR correction were considered significantly enriched.

## Weighted correlation network analysis and enrichment result comparison

We performed weighted correlation network analysis (WGCNA) using gene expression data obtained by two methods respectively to identify DEGs obtained by pairwise comparisons associated with acoustic resting frequency (RF) [31]. We set the optimal the soft thresholding power to 12, the deepSplit value to 2, the minimum tree truncation value to 50 and the height

cut off to 0.25. To better understand gene expression pattern related to phenotypic characteristics, DEGs in modules highly correlated with RF (correlation coefficient higher than 0.8) were selected to perform GO and KEGG enrichment analyses. GO terms and KEGG pathways with FDR value less than 0.05 after FDR correction were considered significantly enriched. We then compared those significantly enriched GO terms and KEGG pathways obtained by the two methods.

### Gene set enrichment analysis

Gene set enrichment analysis (GSEA), which considers the complex network of gene expression, is more likely to detect the effects of subtle but coordinated changes in biological pathways and can avoid ignoring genes that have no obvious differential expression but play an important role in regulating auditory phenotype after screening for DEGs [32–34]. We used the local version of the GSEA v4.2.3 to obtain differentially expressed gene sets by sequencing the expression of all genes in pairwise comparisons (HN vs. JL, HN vs. YN, and YN vs. JL) using reference-based data and observing whether genes in the predefined gene set were enriched at the top or bottom of the sequencing table [35–37]. The *p*-value of enrichment scores and false discovery rate (FDR) of normalization enrichment scores calculated by GSEA were used to identify significantly up-regulated gene sets. Gene sets with a *p*-value less than 0.05 and FDR value less than 0.25 were considered significantly up-regulated.

## Results and discussion

### Acquisition of transcriptome data

After filtering the raw data, more than 95% reads of raw data were retained as clean data, and the error rate of each sample was less than 0.03. The GC content (49.29–53.06%) was not biased. Q20 ranged between 93.48%–95.55% and Q30 ranged between 85.01%–89.31%; these findings indicated that that high-quality clean data were obtained for subsequent analysis (S1 Table). The ratio of clean reads successfully mapping to genomes ranged between 83.51–87.68% (Table 1) after quality control, which indicated that clean reads had a good coverage rate and could be used for subsequent analyses. All details of genes obtained after mapping clean reads to the reference genome are shown in S2 Table.

### Comparison of DEGs obtained by reference-based and reference-free methods

Gene expression pattern in cochlear tissues showed a significant divergence from different geographical populations (Fig 1 and S2 Fig). We obtained a total of 4452 DEGs in the reference-based method, including 3579, 1308, and 1012 DEGs in the comparisons HN vs. JL, HN vs. YN, and YN vs. JL comparisons, respectively (S8 Table), and a total of 18003 DEGs in the reference-free method, including 15484, 2519, and 7468 DEGs in the three comparisons (S9 Table). Both the two methods showed that the HN vs. JL comparison had the most DEGs. Gene expression patterns of HN were more similar to those of YN than JL according to the hierarchical clustering heatmap (Fig 1). But different results of the most same pair were gained using the two methods.

After comparing the DEGs obtained by the two methods, we found 1077 DEGs that were obtained both two methods (Fig 2A, S8 Table). Fewer DEGs were obtained using the reference-based method than the reference-free method, but there were more functionally annotated DEGs using the reference-based method than the reference-free method (Fig 2B, S9, S10 Tables) [22]. DEGs obtained by the reference-based method had a higher annotation rate,

**Table 1.  Mapping statistics of clean reads obtained from 14 samples.**

| Samples | Total reads | Total map (%) | Unique map (%) | Multi map (%) |
|---|---|---|---|---|
| HN1 | 28180774 | 23713205 (84.2%) | 23099619 (82.0%) | 613586 (2.2%) |
| HN2 | 26177304 | 21909565 (83.7%) | 21481834 (82.1%) | 427731 (1.6%) |
| HN3 | 27330508 | 23401123 (85.6%) | 22872788 (83.7%) | 528335 (1.9%) |
| HN4 | 31624162 | 26925007 (85.1%) | 26332588 (83.3%) | 592419 (1.9%) |
| HN5 | 35551624 | 29690270 (83.5%) | 28925851 (81.4%) | 764419 (2.2%) |
| JL1 | 26825364 | 22808700 (85.0%) | 21970001 (81.9%) | 838699 (3.1%) |
| JL2 | 29587584 | 25263212 (85.4%) | 24607119 (83.2%) | 656093 (2.2%) |
| JL3 | 30033230 | 25584917 (85.2%) | 24632113 (82.0%) | 952804 (3.2%) |
| JL4 | 26310962 | 22927800 (87.1%) | 21693410 (82.5%) | 1234390 (4.7%) |
| JL5 | 30469428 | 26714866 (87.7%) | 25958421 (85.2%) | 756445 (2.5%) |
| YN1 | 33796668 | 28783279 (85.2%) | 27591116 (81.6%) | 1192163 (3.5%) |
| YN2 | 28255148 | 24371794 (86.3%) | 23568292 (83.4%) | 803502 (2.8%) |
| YN3 | 25005330 | 20958886 (83.8%) | 20298698 (81.2%) | 660188 (2.6%) |
| YN4 | 25984970 | 22021800 (84.8%) | 21425361 (82.5%) | 596439 (2.3%) |

Total read, the number of clean reads after quality control of raw data; Total map: the number and percentage of reads mapped to the reference genome; Unique map, number and percentage of reads mapped to unique locations on the reference genome (used for subsequent quantitative data analyses); Multi map, number and percentage of reads mapped to multiple locations on the reference genome.

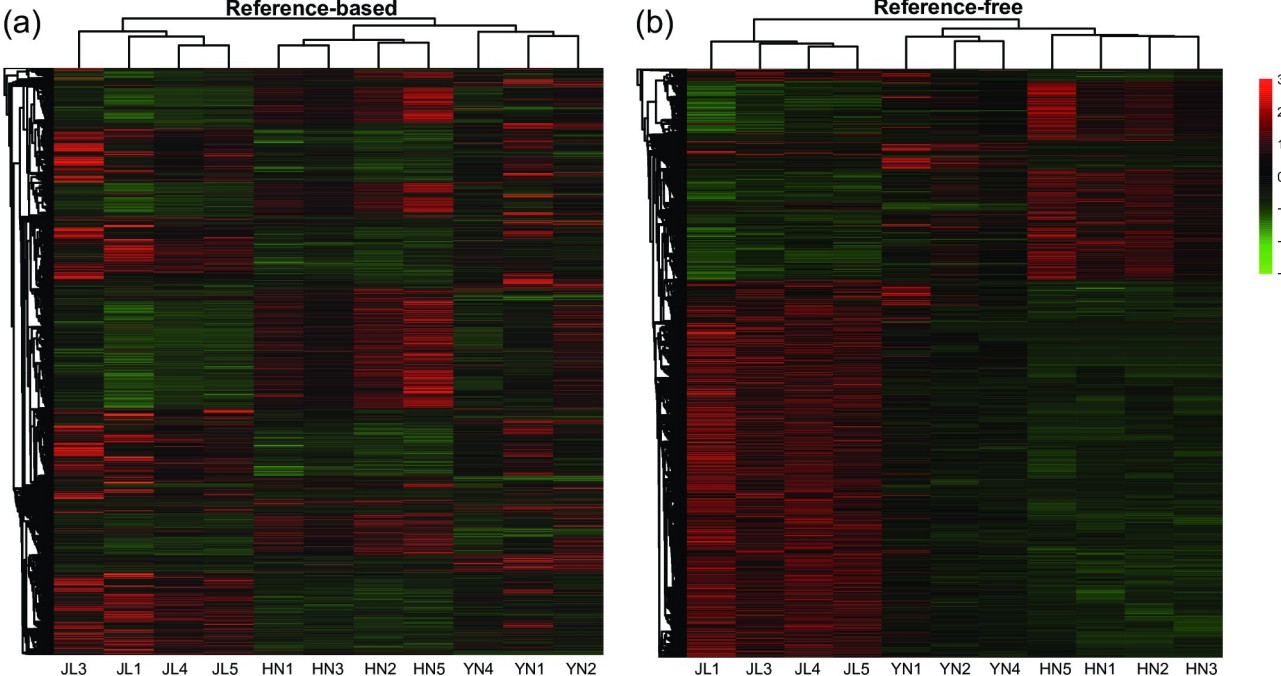

**Fig 1. Clustering results of the remaining samples excluding outliers based on genes obtained by the two methods.** (a) Expression heatmap clustering based on all differentially expressed genes (DEGs) obtained by pairwise comparisons (HN vs. JL, HN vs. YN, and YN vs. JL) in the reference-based method. (b) Expression heatmap clustering based on all DEGs in the reference-free method. Gene expression levels are depicted as standardized (log2-FPKM+1).

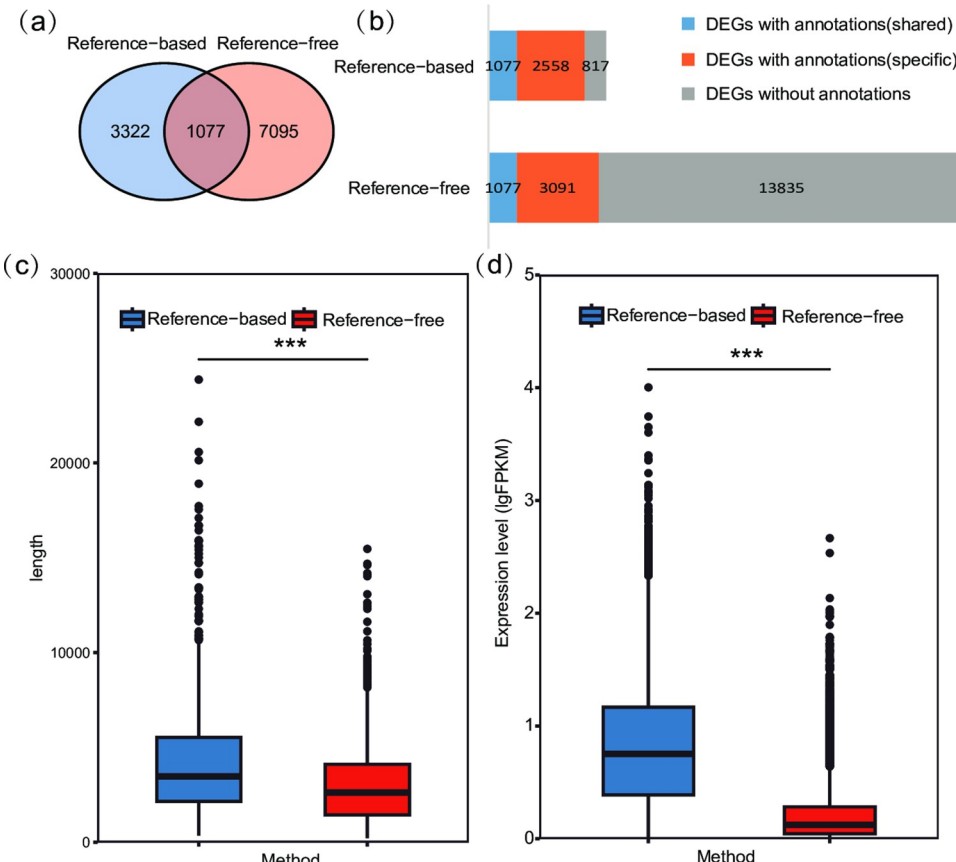

**Fig 2. Comparison of DEGs obtained by pairwise comparisons (HN vs. JL, HN vs. YN, and YN vs. JL) using reference-based method and reference-free methods.** (a)Venn diagram showing the number of DEGs obtained by reference-based and reference-free methods. (b) The annotation results of DEGs obtained by the two methods. The numbers of functionally annotated DEGs that were shared by the two methods, those that were only obtained by one method, and those without annotations are labeled on the histogram plot. (c) Boxplot of shared DEG expression levels (depicted as lg-FPKM+1) obtained by the two methods. (d) Boxplot of shared DEG length obtained by the two methods. '***' were plot because the p-values calculated using paired Mann–Whitney U test were less than 0.001.

which indicated that the reference-free method might obtain DEGs with high false-positive rates, and reference genomes could help increase DEG accuracy and reliability.

The key to subsequent functional analysis is correctly identifying DEGs and accurately assessing gene expression levels, which first requires accurate mapping of RNA sequences to their genomic origins [38, 39]. Although the reference-free method found more DEGs, there were more functionally annotated DEGs obtained by the reference-based method. These findings indicated that there were false-positive results in DEGs obtained by the reference-free method, and this phenomenon will always exist regardless of the assembly tools, parameters, and settings that are used [15, 40–42]. Ockendon et al. [43] compared the transcriptome annotation results of *Drosophila* species using two RNA-Seq methods, and demonstrated that the DEG results obtained by the reference-based method was significantly superior to the results obtained by the reference-free method in terms of both quantity and accuracy.

Zhao [39] found that the reference-free method cannot align long junction reads across introns, especially junction reads spanning more than two exons when eukaryotes were chosen for study. Additionally, although almost all genes spliced in the reference-free method, which were called unigenes, were successfully matched to the reference genome (70275 out of 70704

unigenes), the identified genes and the results of subsequent analyses were different from the results of reference-based method. The mapping result of unigenes assembled by the reference-free method showed that several short unigenes should be identified as one gene (S2 and S11 Tables). Lengths of the shared DEGs sequences obtained by the reference-free method were also significantly shorter than those obtained by the reference-based method (Fig 2C, S8 Table).

The reference-free method has limitations, such as gene identification bias, low transcriptome coverage, and high-false positive rates. These incorrect gene identifications would affect subsequent gene annotation, resulting in errors in functional analysis results. Sequence identification deviations would have a great impact on gene transcript abundance, and eventually lead to the underestimation of transcription levels of some important genes [44, 45]. Our results showed that although gene expression levels of shared DEGs obtained by the two methods were strongly correlated (Fig 2C), significant expression level differences of the same genes between the two methods were shown by paired Mann–Whitney U test (Fig 2D, S8 Table). Lee et al. [12] also found that the reference-free method might underestimate gene expression levels.

However, we found some DEGs involved in hearing processes only obtained by the reference-free method, such as *DFNA5*, *FKBP8* and *POU3F4* [46–50]. These genes indicated that the reference-based method might also have some limitations. First, a single reference genome cannot cover all information of intraspecific variation, which would result in the loss of the genetic information in highly differentiated regions [44, 51]. These regions might play an important role in phenotypic variation and environmental adaptation [11]. At this point, the reference-free method can prevent this situation by obtaining key genes and pathways that cannot be obtained by the reference-base method.

It is worth noting that, even if a gene is confirmed to be related to hearing in one species, it does not necessarily mean that it also plays a role in other species. Hosoya et al. [53] found that *DFNA5* which was believed to be related to human hearing, did not have a similar function in mouse models. As there is no reference genome, annotations of DEGs using the reference-free method need to refer to gene annotations of other species. Therefore, it is important to include validation experiments based on obtained results.

## Functions of shared DEGs obtained by both reference-base and reference-free methods

Although the genes shared by both methods accounted for a small proportion of all DEGs, many genes such as *TMC1*, *TRPC3*, *ASIC1*, *ASIC2*, *SEMA3E*, *CRYM*, *GRHL2*, *COCH*, *WFS1*, *GRM8*, *ANK2*, *SLC16A6*, *ARSG*, and *RIMBP2* might be related to auditory phenotype [52–55]. Then we performed functional enrichment analyses using these shared DEGs and obtained 44 GO terms and 7 KEGG pathways that were significantly enriched (S12 and S13 Tables).

GO analysis results covered three domains of ontology, biological process (BP), cell component (CC), and molecular function (MF), and included terms related to ion channel activity, energy metabolism and nerve conduction process. Additionally, KEGG pathways were related to the nervous system and cellular information transmission process.

The GO terms and KEGG pathways obtained using these shared DEGs were related to ion transport, structure of cell membrane, glutamate receptor activity, and the nervous system, and were found to play important roles in the auditory process of the cochlea. Bats are more likely to pick up high-frequency calls when the cochlea has high voltage, which enhances hearing sensitivity caused by active transport of ions inside and outside of cochlear nerve cells. Additionally, glutamate, as an excitatory neurotransmitter of hair cell synapses, is involved in

the process of listening to signal transmission associated with acoustic stimulation [21, 56–58]. These findings indicated that these genes, which were found to be differentially expressed among populations using both methods, might be significantly associated with phenotypic divergence among populations.

## Comparison of RF-related results obtained by reference-based and reference-free methods

Based on the DEGs obtained by the two methods, we performed WGCNA to construct gene co-expression networks to find DEGs associated with RF phenotype (Fig 3 and S3 Fig). Six modules (including 2544 genes) were found to be significantly correlated with RF phenotype ($p < 0.05$) using the reference-based method DEG results, while eight modules (including 9776 genes) were found using the reference-free DEG results (S15 and S16 Tables).

We further integrated DEGs in RF-related modules obtained by the reference-based method and subsequently performed GO and KEGG enrichment analyses (Fig 4 and S4 Fig). In total, 83 GO terms and 29 KEGG pathways were significantly enriched that were obtained by the reference-based method (S17 and S18 Tables) and were related to transmembrane transport, ion channels and various receptor activities. Alternatively, 97 GO terms and 13 KEGG pathways were obtained by the reference-free method (S19, S20 Tables).

There were several GO terms that were only obtained by the reference-based method, such as the GO terms "inorganic molecular entity transmembrane transporter activity" (GO:0015318, FDR = 4.11E - 08) and "proton transmembrane transport" (GO:1902600, FDR = 0.003), which indicated that inorganic molecules and protons might also play an important role in auditory phenotypic differences. Claire et al. [59] also indicated that the loss of proton NHE1 transmembrane transport activity would cause sensory nerf-related hearing loss in mice. However, the learning pathway was only found using the reference-free method. Additionally, bats have been proved to be one of the few species that is able to learn vocalizations through auditory feedback from others [60–62].

The functional results obtained by the two methods were partially overlapped, and some of the non-overlapping results included pathways describing the same kind of life activity. This indicated that the pathways obtained by the two methods were complementary to each other and revealed different regulatory behaviors of the life activities. For example, the GO term "ionotropic glutamate receptor complex" (GO:0008328) was discovered by the reference-free method and "ionotropic glutamate receptor activity" (GO:0004970) was found by the reference-based method.

## Further phenotypic differentiation analysis

In addition, considering the complex relationship network between genes and the need to try to find more DEGs and functional results, we performed GSEA by pairwise comparisons of the three populations. Ranked gene lists of the pairwise comparison among the three populations are provided in S21–S23 Tables. Significantly up-regulated gene sets were obtained only in the comparisons of HN vs. JL and YN vs. JL. There were no significant results in the comparison of HN vs. YN. All up-regulated gene sets and core genes inside the gene sets are shown in S24 and S25 Tables.

Significantly up-regulated GO terms were entirely different between the results obtained by the reference-based and reference-free methods, which indicated that genes that were not significantly differentially expressed might also play an important role in phenotypic differentiation. However, significantly up-regulated KEGG pathways were all shared and related to RF phenotype, and GSEA obtained similar results to the two transcriptome analysis methods with

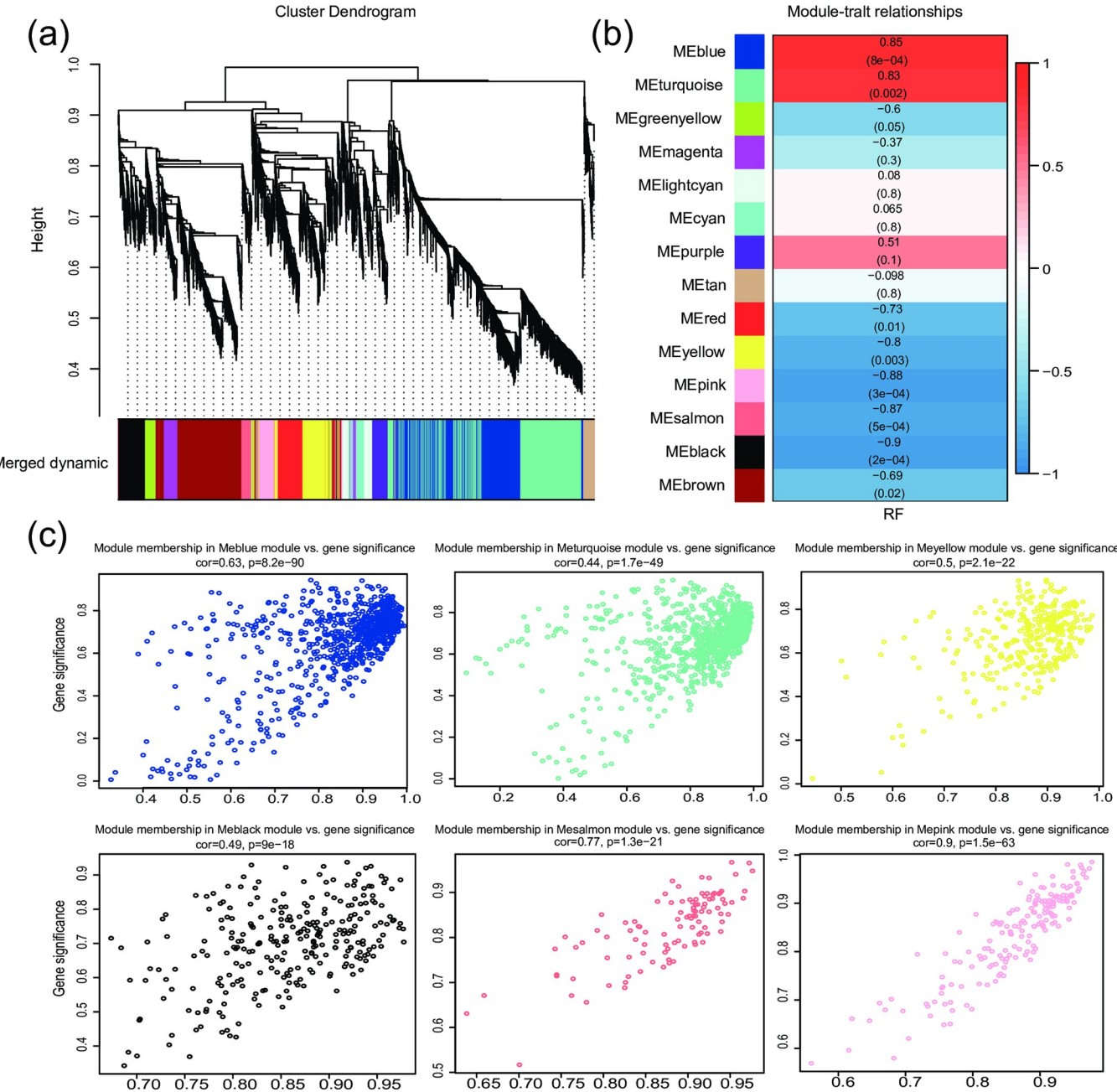

**Fig 3. WGCNA results based on DEGs obtained by pairwise comparisons (HN vs. JL, HN vs. YN, and YN vs. JL) using the reference-based method.** (a) Gene tree spectrum obtained by average linkage hierarchical clustering. (b) Table of module–trait relationships. The correlation coefficient values between the modules and RF phenotype are plotted at the top of each module-trait relationship squares. The *p*-values were labeled under the correlation coefficients in parentheses. (c) Scatter plots showing module membership and gene significance of genes in modules significantly associated with RF phenotypes.

regard to gene function. Significantly up-regulated GO gene sets were related to membrane structure and enzyme activity, and KEGG gene sets were associated with synapses, ion absorption, and neurological diseases (S25 Table). These results indicated that the genes associated with the auditory system are related to transmembrane transport (ion proton inorganic molecules), including membrane structure and ion channel protein (enzyme ion channel protein) activity. Moreover, other key genes were found to be crucial to auditory phenotype

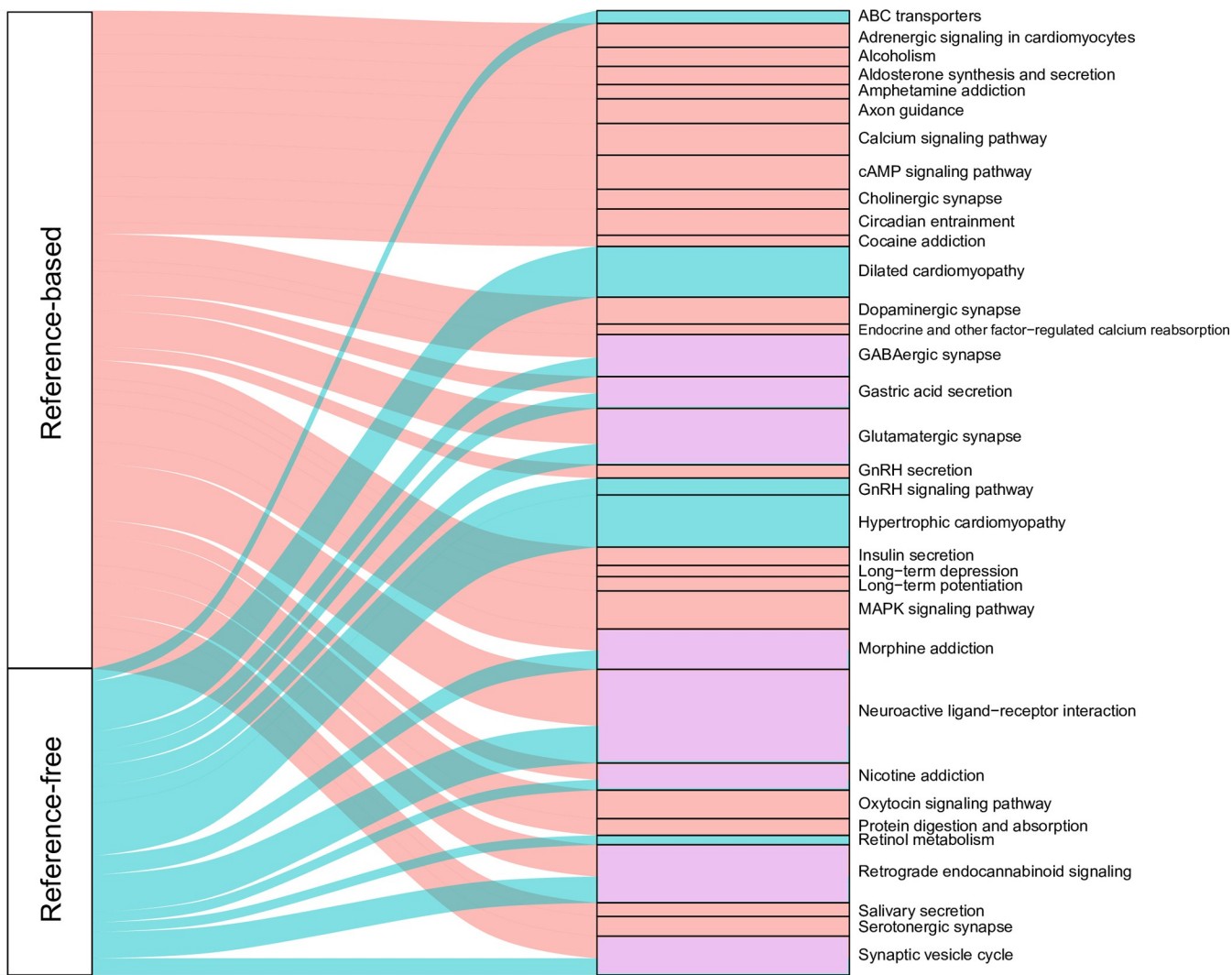

**Fig 4. Sankey diagram showing KEGG pathways using DEGs obtained by the two methods in RF-related modules from WGCNA.** Pathways obtained only by the reference-based method or the reference-free method were colored red and blue, respectively. Pathways obtained by both methods were colored purple. The width of the rectangle shape represents the number of gene counts enriched in the pathways.

differentiation in addition to DEGs, such as *GNG13*, *RGS7*, and *GNG3*, which are all related to guanine nucleotide-binding protein (G protein); which is responsible for the initiating and regulating of transmembrane signaling system [63, 64].

## Conclusions

We performed reference-based transcriptome analysis using RNA data of three horseshoe bat geographic populations in China, and compare the results of differential expression analyses among the three populations and the results related to RF phenotypic differentiation with those of the reference-free method. We also performed GSEA to find more core genes and functions that are important in phenotypic differentiation. We found that the use of reference genomes can help improve the accuracy and reliability of identified DEGs and subsequent functional analyses, reducing the workload increased by fuzzy or ambiguously identified reads; however, the reference-free method can find more possible DEGs that may be distributed in highly differentiated

gene regions of the species' genome that are missed by the reference-based method. Either approach can achieve important results that the other cannot. Thus, it is better to combine the results obtained by the two methods when performing transcriptome analyses and discussing associated results to produce more accurate and comprehensive results.

## Supporting information

**S1 Table. Sequencing data quality statistics using the reference-based method.**
(XLSX)

**S2 Table. Gene details of 14 individuals obtained by the reference-based method.**
(XLSX)

**S3 Table. Gene details of 14 individuals obtained by the reference-free method.**
(XLSX)

**S4 Table. Principal component values of all 14 individuals according to the expression levels of all genes (FPKM) obtained in the reference-based method.**
(XLSX)

**S5 Table. Principal component values of 11 individuals excluding the three outlier samples according to the expression levels of all genes (FPKM) obtained in the reference-based method.**
(XLSX)

**S6 Table. Principal component values of all 14 individuals according to the expression levels of all genes (FPKM) obtained in the reference-free method.**
(XLSX)

**S7 Table. Principal component values of 11 individuals excluding the three outlier samples according to the expression levels of all genes (FPKM) obtained in the reference-free method.**
(XLSX)

**S8 Table. Annotated DEGs obtained by both reference-based and reference-free method.**
(XLSX)

**S9 Table. DEGs obtained by pairwise comparisons of the three populations in the reference-based method.**
(XLSX)

**S10 Table. DEGs obtained by pairwise comparisons of the three populations in the reference-free method.**
(XLSX)

**S11 Table. Mapping locations statistics of unigenes on the reference genome.** sseq_chrid represents the ID of chromosome which the unigenes were mapped to. 'sstart' and 'send' represent the locations of start and end bases on the mapped chromosome, respectively. 'qstart' and 'qend' represent the locations of start and end bases, respectively, of the unigenes mapped to the chromosome.
(XLSX)

**S12 Table. Complete results of the GO enrichment analysis for the genes shared by the two methods.**
(XLSX)

**S13 Table. Complete results of the KEGG enrichment analysis for the genes shared by the two methods.**
(XLSX)

**S14 Table. Resting frequency of 11 individuals excluding the three outlier samples.**
(XLSX)

**S15 Table. DEGs in modules significantly associated with RF by the reference-based method.**
(XLSX)

**S16 Table. DEGs in modules significantly associated with RF by the reference-free method.**
(XLSX)

**S17 Table. Complete results of the GO enrichment analysis for the merged gene sets in modules significantly associated with RF by the reference-based method.**
(XLSX)

**S18 Table. Complete results of the KEGG enrichment analysis for the merged gene set in modules significantly associated with RF by the reference-based method.**
(XLSX)

**S19 Table. Complete results of the GO enrichment analysis for the merged gene sets in modules significantly associated with RF by the reference-free method.**
(XLSX)

**S20 Table. Complete results of the KEGG enrichment analysis for the merged gene set in modules significantly associated with RF by the reference-free method.**
(XLSX)

**S21 Table. GSEA ranked gene list for HN vs. JL.**
(XLSX)

**S22 Table. GSEA ranked gene list for HN vs. YN.**
(XLSX)

**S23 Table. GSEA ranked gene list for YN vs. JL.**
(XLSX)

**S24 Table. Details of genes in significantly up-regulated gene sets.**
(XLSX)

**S25 Table. GO and KEGG enrichment results of significantly up-regulated gene sets.**
(XLSX)

**S1 Fig. Scree plots of the principal component analysis based on the two methods.** Scree plot of all 14 individuals based on the reference-based method (a) and the reference-free method (b). Scree plot of 11 individuals excluding three outlier samples based on the reference-based method (c) and the reference-free method (d).
(TIF)

**S2 Fig. PCA clustering results based on the two methods.** PCA plot of all 14 individuals based on the reference-based method (a) and the reference-free method (b). PCA plot of 11 individuals excluding three outlier samples based on the reference-based method (c) and the reference-free method (d).
(TIF)

**S3 Fig. WGCNA results based on DEGs obtained by pairwise comparisons (HN vs. JL, HN vs. YN, and YN vs. JL) using the reference-free method.** (a) Gene tree spectrum obtained by average linkage hierarchical clustering. (b) Table of module–trait relationships. The correlation coefficient values between the modules and RF phenotype are plotted at the top of each module-trait relationship squares. The *p*-values were labeled under the correlation coefficients in parentheses. (c) Scatter plots showing module membership and gene significance of genes in modules significantly associated with RF phenotypes.
(TIF)

**S4 Fig. Sankey diagram showing shared GO terms using DEGs obtained by the two methods in RF-related modules from WGCNA.** Terms obtained by both methods were colored. The width of the rectangle shape represents the number of gene counts enriched in the terms.
(TIF)

## Acknowledgments

We obtained valuable information from the National Center for Biotechnology Information database (https://www.ncbi.nlm.nih.gov/) for providing valuable information. We thank Mallory Eckstut, PhD, from Liwen Bianji (Edanz) (www.liwenbianji.cn) for editing the English text of a draft of this manuscript. And we also thank Muisha B Mbikyo for his help with the language modification.

## Author Contributions

**Conceptualization:** Xiaoxiao Shi, Jun Li, Tong Liu, Keping Sun, Jiang Feng.

**Formal analysis:** Xiaoxiao Shi, Tong Liu.

**Funding acquisition:** Jiang Feng.

**Investigation:** Hanbo Zhao.

**Methodology:** Xiaoxiao Shi, Jun Li.

**Project administration:** Keping Sun, Jiang Feng.

**Resources:** Hanbo Zhao, Keping Sun, Jiang Feng.

**Supervision:** Jun Li.

**Validation:** Jun Li, Tong Liu.

**Visualization:** Xiaoxiao Shi.

**Writing – original draft:** Xiaoxiao Shi.

**Writing – review & editing:** Xiaoxiao Shi, Jun Li, Haixia Leng, Keping Sun, Jiang Feng.

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
