## [Decision Letter · Decision Letter 0]

14 Dec 2022

PONE-D-22-32713Divergence of cochlear transcriptomics between reference‑based and reference‑free transcriptome analyses among Rhinolophus ferrumequinum populationsPLOS ONE

Dear Dr. Sun,

Thank you for submitting your manuscript to PLOS ONE. After careful consideration, we feel that it has merit but does not fully meet PLOS ONE’s publication criteria as it currently stands. Therefore, we invite you to submit a revised version of the manuscript that addresses the points raised during the review process.

We look forward to receiving your revised manuscript.

Kind regards,

Shailender Kumar Verma, Ph.D.

Academic Editor

PLOS ONE

Journal Requirements:

"This work was supported by the National Natural Science Foundation of China (Grant Nos. 32171525 and 31961123001) and Jilin Provincial Natural Science Foundation (Grant No. 20220101291JC)."

"The authors declare that there is no conflict of interest."

5. We noted in your submission details that a portion of your manuscript may have been presented or published elsewhere. [Although we  used some reference - free transcriptome data which had been published , our study focused on describing the differences between the results of two different methods. What's more,we had got the permission from the  authors of the data.] Please clarify whether this [conference proceeding or publication] was peer-reviewed and formally published. If this work was previously peer-reviewed and published, in the cover letter please provide the reason that this work does not constitute dual publication and should be included in the current manuscript.

7. PLOS requires an ORCID iD for the corresponding author in Editorial Manager on papers submitted after December 6th, 2016. Please ensure that you have an ORCID iD and that it is validated in Editorial Manager. To do this, go to ‘Update my Information’ (in the upper left-hand corner of the main menu), and click on the Fetch/Validate link next to the ORCID field. This will take you to the ORCID site and allow you to create a new iD or authenticate a pre-existing iD in Editorial Manager. Please see the following video for instructions on linking an ORCID iD to your Editorial Manager account: https://www.youtube.com/watch?v=_xcclfuvtxQ

8. Please upload a copy of Supporting Information Table. S2-S11 Table which you refer to in your text on page 26.

Reviewers' comments:

Reviewer's Responses to Questions

**Comments to the Author**

1. Is the manuscript technically sound, and do the data support the conclusions?

Reviewer #1: Yes

Reviewer #2: Yes

2. Has the statistical analysis been performed appropriately and rigorously? 

Reviewer #1: Yes

Reviewer #2: Yes

3. Have the authors made all data underlying the findings in their manuscript fully available?

Reviewer #1: Yes

Reviewer #2: Yes

4. Is the manuscript presented in an intelligible fashion and written in standard English?

Reviewer #1: Yes

Reviewer #2: Yes

5. Review Comments to the Author

Reviewer #1: In the manuscript "Divergence of cochlear transcriptomics between reference‐ based and reference‐ free transcriptome analyses among Rhinolophus ferrumequinum populations", the authors used public reference genome and public RNA-seq dataset of a bat species to compare the performance of reference-based and reference-free methods of RNA-seq data analysis. While I think an in-depth comparison of the two RNA-seq data analysis strategies are definitely worth examining, I also feel that the manuscript appears a bit superficial regarding such comparison in its current status. For example, the authors mentioned that the idea that reference assembly and annotation quality as well as the population-wide polymorphism may affect the performance of the reference-based method but they did not quantitatively test and define such influences using their datasets. I think the manuscript could be significantly improved with such strict technical characterization. The current manuscript reads too descriptive to me especially consider the fact that the RNA-seq data is published data and therefore a general description of the DEG results based on such data might not be innovative enough on its own. Also, the language of this manuscript needs to be substantially improved.

A few minor points are further attached:

page 8 line 27-29: "In this study, we compared the cochlear transcriptome analysis results of Rhinolophus ferrumequinum from three lineages". The authors might want to briefly introduce "Rhinolophus ferrumequinum" with its common name.

page 8 line 31-37: The authors might want to consider improve this section to better reflect the difference between the reference-based and reference-free methods.

page 9 line 52: "qualified" -> "quantified"?

page 9 line 58-62: This sentence needs a rephrasing.

page 10 line 82-83: "which provided a good database for transcriptome analysis with reference genome." => "which provided a good reference genome for transcriptome analysis."

page 10 line 95: "Row data" => "Raw data"

page 12 line 106: Please provide the name, version, and command of the corresponding software used for contamination filtering and adapter trimming.

page 18 line 220: What did the authors mean by "gene reading errors"?

page 25 line 369: What did the authors mean by "obtain more results"?

Reviewer #2: （1）line 123-124 :should be three populations? or rewrite this sentence more accurately.

（2））Figure1:

1.please explain why sample HN4/JL2/YN3 are not included in the gene clustering result?

2.if you excluded these three samples, have you excluded them in the subsequent analysis?

（3））Figure2: not labeled

2 types of analytical method: reference-based and reference-free

1. How much do the differentially expressed genes obtained by each analytical means differ between different samples of the same species of bats?

2. Comparison of differentially genes between the two analytical means for the same species of bats.

3. Comparison of differentially expressed genes between bats of different species by the same method.

6. PLOS authors have the option to publish the peer review history of their article (what does this mean?). If published, this will include your full peer review and any attached files.

Reviewer #1: No

Reviewer #2: No

---

## [Author Response · Author response to Decision Letter 0]

20 Feb 2023

We are very appreciative of the constructive comments on the manuscript from the reviewers. We have carefully and seriously revised the manuscript according to the comments. These comments are very helpful for us to revise and improve our manuscript.

For the comment 1 from the editor, we changed the following parts in the manuscript to meet PLOS ONE's style requirements.

1. We adjusted the format of the first page, and added lacking Author Byline (please see page 1).

2. We standardized the format of Table 1 (please see page 11 and 12).

3. We added page number under each page.

4. We deleted all unpublished contents in the reference. 

5. We renamed all files as required.

For the comment 2 from the editor, we have checked the grant numbers for the awards when we resubmit the manuscript.

For the comment 3 from the editor, we have checked the grant numbers for the awards when we resubmit the manuscript. We ask for changing the financial disclosure to the following statement.

This work was supported by grant nos. 32171525 and 31961123001 from the National Natural Science Foundation of China (to KS and JF), and grant 20220101291JC from the Natural Science Foundation of Jilin Province (to KS). The funders had no role in study design, data collection and analysis, decision to publish, or preparation of the manuscript.We have included this statement in the new cover letter.

For the comment 4 from the editor, we have included the statement “The authors have declared that no competing interests exist.” in the new cover letter.

For the comment 5 from the editor, we clarify that the original data we used and results of reference-free method described in the manuscript has been published under peer review, but we have quoted and integrated this part with our new reference-base analysis results to obtain the difference between the results of two methods, so it does not constitute a duplicate publication.

We have included this statement in the new cover letter.

For the comment 6 from the editor, we have added all underlying data to the supporting information of the manuscript.

For the comment 7 from the editor, we had added the ORCID iD of the corresponding author.

For the comment 8 from the editor, because we changed several parts in the manuscript, the contents of Supporting Information had been changed. Now we reuploaded all Supporting Information Tables.

For the comment 1 from reviewer 1, many thanks for those constructive suggestion, and we fully agree with your comments. According to your comments, we have made the following changes:

1. We have deleted or refined those long-winded and superficial descriptions. And to make the results more qualified, we added several analyses.

2. As we focus on the difference of results in subsequent transcriptome analyses, it’s hard for us to compare two methods in technical level. So we quoted articles focus on this field and disgusted using their research results.

The idea that the results obtained by reference‑based method could be affected by the accuracy and completeness of the reference genome and the idea that the population-wide polymorphism may affect the performance of the reference-based method were quoted from several researches.

3. We have deleted unnecessary descriptions of the reference-free results [6] and refined those long-winded and superficial descriptions in the “Results and discussion” section.

4. We are sorry for the language problem, and the new manuscript has been carefully edited by a professional language editing service to improve the grammar and readability.

For the comment 2-7 from reviewer 1, we have corrected all spelling mistakes.

For the comment 8 from reviewer 1, we stated the name, version, and command of the corresponding software used for contamination filtering and adapter trimming in the manuscript.

For the comment 9-10 from reviewer 1, we explained the meaning of the phrases and rewrote those sentences to make them more accurate and understandable.

For the comment 1 from reviewer 2, we rewrote those sentences to make those ambiguous expressions clearer.

For the comment 2 from reviewer 2, changed the contents and legends of Figure 2 to make the figure easier to understand. We also added more analyses to further compare the difference of DEGs between two methods.

---

## [Decision Letter · Decision Letter 1]

16 Mar 2023

PONE-D-22-32713R1Divergence of cochlear transcriptomics between reference‑based and reference‑free transcriptome analyses among *Rhinolophus ferrumequinum* populationsPLOS ONE

Dear Dr. Sun,

Thank you for submitting your manuscript to PLOS ONE. After careful consideration, we feel that it has merit but does not fully meet PLOS ONE’s publication criteria as it currently stands. Therefore, we invite you to submit a revised version of the manuscript that addresses the points raised during the review process.

We look forward to receiving your revised manuscript.

Kind regards,

Shailender Kumar Verma, Ph.D.

Academic Editor

PLOS ONE

Reviewers' comments:

Reviewer's Responses to Questions

**Comments to the Author**

1. If the authors have adequately addressed your comments raised in a previous round of review and you feel that this manuscript is now acceptable for publication, you may indicate that here to bypass the “Comments to the Author” section, enter your conflict of interest statement in the “Confidential to Editor” section, and submit your "Accept" recommendation.

Reviewer #1: All comments have been addressed

Reviewer #2: All comments have been addressed

2. Is the manuscript technically sound, and do the data support the conclusions?

Reviewer #1: Yes

Reviewer #2: Yes

3. Has the statistical analysis been performed appropriately and rigorously? 

Reviewer #1: Yes

Reviewer #2: N/A

4. Have the authors made all data underlying the findings in their manuscript fully available?

Reviewer #1: Yes

Reviewer #2: Yes

5. Is the manuscript presented in an intelligible fashion and written in standard English?

Reviewer #1: Yes

Reviewer #2: Yes

6. Review Comments to the Author

Reviewer #1: The revised manuscript has been largely improved in terms of structure and readability. Most of my previous comments have been addressed by the authors. However, if possible, I would like to ask the authors to make additional efforts (see the major concern below) as I feel this is important.

Major concern:

The reference-based method implemented in this study uses FPKM for transcript quantification, whereas the reference-free method implemented in the cited study (Zhao et al. 2019) uses RPKM instead. Therefore, it is difficult to conclude how much of the observed DEG difference is due to the FPKM vs RPKM contrast and how much of it is due to the reference-based vs. reference-free contrast. Therefore, I would recommend the authors to re-run the reference-free analysis with FPKM for a proper comparison, especially for those analysis related to expression level quantification.

Minor concerns/suggestions:

page 4 line 59: maybe deleting "splice" at this line, otherwise it doesn't make much sense.

page 5 line 77: "basic" => "generic"

page 6 line 100-108: Please provide the URL (or accession number) for the reference genome/transcriptome/annotation used.

page 11 line 192-196: Please state explicitly that the presented results in this paragraph are based on reference-based method.

page 12 line 202: "more annotated DEGs" => "more functionally annotated DEGs" . Same correction applies to Fig2 as well.

page 14 line 244: Better to give a formal definition for "unigenes" here before referring to it.

page 15 line 272: "the other" => "other"

page 18: To show the GO term and KEGG pathways differed between the reference-based and reference-free methods, maybe the authors could consider using parallel set plots (https://ggforce.data-imaginist.com/reference/geom_parallel_sets.html) for visualization.

page 20: "more genes were found": "more" relative to what? Please clarify the comparison intended here.

Reviewer #2: (No Response)

7. PLOS authors have the option to publish the peer review history of their article (what does this mean?). If published, this will include your full peer review and any attached files.

Reviewer #1: No

Reviewer #2: **Yes: **Yao Xinsheng

---

## [Author Response · Author response to Decision Letter 1]

12 Jun 2023

Thanks for all of your valuable comments. According to your new suggestions, we calculated the FPKM of genes gained in the reference-free method and redid all subsequent reference-free analyses with FPKM and all comparisons related to expression level quantification. We have also corrected the grammatical or spelling errors you raised. All changes in the new manuscript have been highlighted.

---

## [Decision Letter · Decision Letter 2]

26 Jun 2023

Divergence of cochlear transcriptomics between reference‑based and reference‑free transcriptome analyses among *Rhinolophus ferrumequinum* populations

PONE-D-22-32713R2

Dear Dr. Sun,

We’re pleased to inform you that your manuscript has been judged scientifically suitable for publication and will be formally accepted for publication once it meets all outstanding technical requirements.

Kind regards,

Shailender Kumar Verma, Ph.D.

Academic Editor

PLOS ONE

Additional Editor Comments (optional):

One reviewer has already accepted the first draft of the manuscript while another reviewer endorsed the revised version. Based on the recommendations of both Reviewers, I also recommend accepting the manuscript. 

Reviewers' comments:

Reviewer's Responses to Questions

**Comments to the Author**

1. If the authors have adequately addressed your comments raised in a previous round of review and you feel that this manuscript is now acceptable for publication, you may indicate that here to bypass the “Comments to the Author” section, enter your conflict of interest statement in the “Confidential to Editor” section, and submit your "Accept" recommendation.

Reviewer #1: All comments have been addressed

2. Is the manuscript technically sound, and do the data support the conclusions?

Reviewer #1: Yes

3. Has the statistical analysis been performed appropriately and rigorously? 

Reviewer #1: Yes

4. Have the authors made all data underlying the findings in their manuscript fully available?

Reviewer #1: Yes

5. Is the manuscript presented in an intelligible fashion and written in standard English?

Reviewer #1: No

6. Review Comments to the Author

Reviewer #1: I appreciate the authors' efforts during the revision and I think the manuscript has been much improved.

7. PLOS authors have the option to publish the peer review history of their article (what does this mean?). If published, this will include your full peer review and any attached files.

Reviewer #1: No

---

## [Editor Report · Acceptance letter]

3 Jul 2023

PONE-D-22-32713R2 

Divergence of cochlear transcriptomics between reference‑based and reference‑free transcriptome analyses among *Rhinolophus ferrumequinum* populations 

Dear Dr. Sun:

I'm pleased to inform you that your manuscript has been deemed suitable for publication in PLOS ONE. Congratulations! Your manuscript is now with our production department. 

Kind regards, 

on behalf of

Dr. Shailender Kumar Verma 

Academic Editor

PLOS ONE